# Improved Survival Analyses Based on Characterized Time-Dependent Covariates to Predict Individual Chronic Kidney Disease Progression

**DOI:** 10.3390/biomedicines11061664

**Published:** 2023-06-08

**Authors:** Chen-Mao Liao, Chuan-Tsung Su, Hao-Che Huang, Chih-Ming Lin

**Affiliations:** 1Department of Applied Statistics and Information Science, Ming Chuan University, Taoyuan 333, Taiwan; cmliao@mail.mcu.edu.tw (C.-M.L.);; 2Department of Healthcare Information and Management, Ming Chuan University, Taoyuan 333, Taiwan; ctsu@mail.mcu.edu.tw

**Keywords:** artificial neural network, chronic kidney disease, Kaplan–Meier, random survival forest

## Abstract

Kidney diseases can cause severe morbidity, mortality, and health burden. Determining the risk factors associated with kidney damage and deterioration has become a priority for the prevention and treatment of kidney disease. This study followed 497 patients with stage 3–5 chronic kidney disease (CKD) who were treated at the ward of Taipei Veterans General Hospital from January 2006 to 2019 in Taiwan. The patients underwent 3-year-long follow-up sessions for clinical measurements, which occurred every 3 months. Three time-dependent survival models, namely the Cox proportional hazard model (Cox PHM), random survival forest (RSF), and an artificial neural network (ANN), were used to process patient demographics and laboratory data for predicting progression to renal failure, and important features for optimal prediction were evaluated. The individual prediction of CKD progression was validated using the Kaplan–Meier estimation method, based on patients’ true outcomes during and beyond the study period. The results showed that the average concordance indexes for the cross-validation of the Cox PHM, ANN, and RSF models were 0.71, 0.72, and 0.89, respectively. RSF had the best predictive performances for CKD patients within the 3 years of follow-up sessions, with a sensitivity of 0.79 and specificity of 0.88. Creatinine, age, estimated glomerular filtration rate, and urine protein to creatinine ratio were useful factors for predicting the progression of CKD patients in the RSF model. These results may be helpful for instantaneous risk prediction at each follow-up session for CKD patients.

## 1. Introduction

Chronic kidney disease (CKD) is a significant global health problem and a common health issue. The prevalence is increasing worldwide, amounting to more than 800 million patients [1]. CKD is considered a high-risk clinical disease with frequent adverse events [2] and is associated with significant mortality [3]. The ratio of prevalence and incidence of CKD in Taiwan is relatively high compared with other countries [4]. Additionally, numerous complications of CKD have been observed. For example, hypertension [5], cardiovascular disease (CVD) [6], and diabetes [7], all of which are recognized as strong risk factors for renal disease. Importantly, the progression of CKD may be correlated with the individual risk factors of each patient, and early identification and accurate prognostication may help clarify the natural history of CKD progression.

The estimated glomerular filtration rate (eGFR) is one of the most important risk factors for identifying the classification of CKD [8]. A lower eGFR represents progressively more severe stages of CKD and eventually leads to end-stage renal disease (ESRD). Tangri et al. concluded that eGFR is a time-dependent predictor that can improve the risk prediction of CKD progression [9]. Additionally, the magnitude of proteinuria is an important marker of prognosis in CKD [10]. Hoy et al. showed progressively lower eGFRs in people with increasing intensities of pathologic albuminuria [11]. Raised urinary albumin excretion was associated with increased renal and cardiovascular mortality in a remote Australian aboriginal community [12]. Another study showed that serum albumin, serum creatinine, albumin/creatinine ratio, and hemoglobin are multivariate risk factors for ESRD, which were taken from 1513 subjects included in the reduction of endpoints in noninsulin-dependent diabetes with the Angiotensin II Antagonist Losartan study [13]. In our recent study, using the Shapley additive explanation value method, the urine creatinine and eGFR were the most and second-most important predictive features in patients diagnosed with advanced-stage CKD within 3 and 5 years. In addition, serum creatinine was the most important predictive feature in patients diagnosed with advanced-stage CKD within 1–3 years [14]. Ultimately, the key predictive features may help determine the optimal predictive models for the progression from CKD to ESRD.

The prediction of CKD progression is an important task for patient care in clinical management. Machine learning (ML) methods have been used to predict the risk of CKD applications in recent years [15,16,17,18]. In addition, several risk prediction models have been proposed for CKD applications [14,19,20,21,22]. The published models applied to renal diseases include the Cox proportional hazard model (Cox PHM) [19,23], random survival forest (RSF) [24], and artificial neural network (ANN) models [22,25]. Cox PHM is the most widely used method to predict the risk factors of clinical diseases in cohort studies. Cox PHM can be used to examine the covariate effects on the hazard function to determine the failure time variable. However, it may not fit the data well due to several limitations, such as its reliance on restrictive assumptions such as the proportionality of hazards and linearity [26]. Therefore, there is an increased risk of overfitting, which diminishes the statistical power of the model. In our recent study, using baseline data, five available classification models (i.e., Gaussian naïve Bayes, linear regression, random forest (RF), support vector machine, and extreme gradient boosting) were developed for predicting the risk of progression among patients with CKD. The results showed that the RF model demonstrated the highest performance compared to the other models [14]. RF can be adapted to handle complex survival data, including nonlinear effects and complex interactions between features, which may be inappropriate for conventional statistical models. RSF is a nonparametric method that generates multiple decision trees to analyze right-censored survival data [27]. A cumulative hazard function (CHF) can be generated from each decision tree, which is averaged into an ensemble CHF. RSF has demonstrated better performance compared to Cox PHM based on the prediction error criterion [28], and it has been used for clinical applications, such as tumor and incident risk of diabetes [29,30]. On the other hand, the deep neural networks have received considerable attention for predicting the occurrence of events of interest, especially for right-censored time-to-event data. Recently, Zhao and Feng noted that DNNSurv is capable of predicting the conditional survival probability in each interval, and the marginal survival probability can be used to evaluate the discrete-time survival framework to predict both the marginal and the conditional survival probabilities, or the complementary risks [31]. Several neural network models, including Cox-nnet, DeepSurv, nnet-survival, and DNNSurv, have been utilized in survival analyses and have undergone peer review. These models also have well-documented source codes. Among them, DNNSurv and nnet-survival do not rely on the proportional hazard assumption, which is less questionable when the number of covariates is large. In comparison to nnet-survival, DNNSurv employs a theoretically justified pseudo-value approach to handle censored data, irrespective of whether the censoring occurs in the first or second half of the interval. Furthermore, DNNSurv can handle data with covariate-dependent censoring, a capability lacking in nnet-survival. Additionally, DNNSurv is specifically designed to circumvent the sophisticated network structure introduced by censored data, which is required by deep neural network models such as convolutional or recurrent neural networks [31]. The Cox-based DNNSurv model was selected as an ANN method for comparison among models in our study.

The time-to-event data in CKD typically refers to the duration it takes for a specific event to occur, such as ESRD or death. By incorporating right-censored time-to-event data into the analysis, the risk factors of CKD progression can be evaluated for the varying follow-up times of patients. This allows for a more comprehensive assessment of the disease’s progression and a better understanding of the risk factors that influence the timing of events in CKD patients. Accurate risk prediction models can inform clinical decision-making and help identify patients who are at a higher risk of adverse outcomes, such as ESRD or mortality. Thus, a suitable model with a time-dependent covariate for predicting CKD progression is required. However, such a model has yet to be developed and established.

In the present study, the performances of the Cox PHM, RSF, and ANN models with time-dependent variables were used to determine the risk of CKD progression over 3 years. Furthermore, the important features were evaluated as high-risk factors for determining the optimal predictive features for the models. The integration of time-dependent covariates into survival analysis using ML methods for predicting CKD progression was investigated. By characterizing and considering time-dependent covariates, the proposed methodology can predict the instantaneous risk prediction at each follow-up session for CKD patients and provide a more comprehensive understanding of the factors influencing disease progression. This novel approach has the potential to improve the precision of prognostic predictions and could assist in developing tailored interventions and treatment strategies to better manage CKD.

## 2. Materials and Methods

### 2.1. Patient Population and Study Design

A retrospective cohort study using de-identified pathological records was conducted in this study. The dataset was collected from November 2006 to December 2019 in a branch of the Taipei Veterans General Hospital. A total of 947 patients were collected from the National Health Insurance (NHI) CKD program, which is a clinical care and education program for patients initiated by the NHI administration under the Ministry of Health and Welfare in Taiwan. This study was approved by the Institutional Review Board of the Taipei Veterans General Hospital (No. 2020-01-024BC).

In the data, the patients who had been diagnosed with stages 1 and 2 were excluded. Similarly, patients with insufficient follow-up data (less than 3 clinical visits) were also excluded. Missing values in patients with insufficient pathological records were replaced with multiple imputations [32]. Data imputation was performed using the multivariate imputation method and chained equations module in the R package. Then, the patients diagnosed with stages 3–5 were investigated (N = 497), focusing on patients who progressed from CKD to ESRD within 3 years. This study included 352 patients in stage 3, 69 patients in stage 4, and 76 patients in stage 5. The number of patients with stage 3 who progressed to dialysis was 22. In addition, the number of patients with stages 4 and 5 that progressed to dialysis was 6 and 40, respectively. A flow chart depicting the patient selection and categorization processes is shown in Figure 1.

The characteristics of pathological records for each patient were determined during the prespecified 90-day assessment period, starting from the first clinic visit until the study endpoint date. The baseline characteristics of pathological records were defined as an initial point. The endpoint was defined as the requirement for dialysis or kidney transplantation. The patients were followed up until the endpoint date, when patients with dialysis or renal failure were established during the observation period.

The study investigated the time-dependent predictive risk factors of CKD patients with stage 3–5 progression. The eGFR, serum creatinine, natural logarithm-transformed urine protein to serum creatinine ratio (PCRln), and glycated hemoglobin (HbA1c) were used as time-dependent predictors for data characterization. To further characterize the differences among the risk factors of CKD, the time-dependent variability analysis $\gamma \left(i\right)$ was used, which is defined as the predictor variables divided by the observation period and is represented as follows:(1)$$\gamma \left(i\right)=\left\{\begin{array}{cc} \frac{v_{i+1}-v_{i}}{t_{i+1}-t_{i}}, & i=1,2\\ \frac{\sum_{j\in R_{i}}\left(v_{j}-{\bar{v}}_{i}\right)\left(t_{j}-{\bar{t}}_{i}\right)}{\sum_{j\in R_{i}}{\left(t_{j}-{\bar{t}}_{i}\right)}^{2}}, & i=3,4,\cdots ,f \end{array}\right.\;,\;\;{\bar{v}}_{i}=\frac{\sum_{j\in R_{i}}v_{j}}{i},\;{\bar{t}}_{i}=\frac{\sum_{j\in R_{i}}t_{j}}{i},\;R_{i}=\left\{j|t_{j}\leq t_{i}\right\},\;$$
where f is the number of patient clinic visits, $v_{1},\;v_{2},\cdots ,v_{f}$ is each measurement of risk factor, $t_{i}$ is the observation period, and ${\bar{v}}_{i}$ is the mean value of risk factors during the previous *i* period. In addition, the regression analysis was performed to determine the relationship between the predictor variables and observation period for *i* ≥ 3.

To assess the high-risk factors in predictive models for progression from CKD to ESRD, the important characterized factors of CKD were selected by multivariate analysis of variance (MANOVA) and the independent chi-squared test. Randomized data subsets were used for cross-validation (K = 5). The Cox PHM, RSF, and ANN were used to determine the time-dependent prediction of CKD progression. Then, a comparison of the concordance index (C-index) and Kaplan–Meier method (KM) was used to predict the risk of progression to eventual ESRD among CKD patients with stages 3–5. The models were used to identify risk factors for predicting disease progression in CKD within 3 years. The predictor variables of Cox PHM were normalized by z-score transformation, and the value important (VIMP) of RSF was used for feature selection and prediction. The flow chart of model training and performance evaluation is shown in Figure 2.

### 2.2. Mathematical Modeling

In this study, the proportional hazards regression with time-dependent covariates was evaluated. The conditional-hazard function is shown below [33]:(2)$$\lambda \left(t\;|\;\bar{v}\right)=\underset{\Delta t\to 0}{\lim}P(t<T\leq t+\Delta t\;|T\;>\;t,\;\bar{v}\left(t\right))/\Delta t$$
where *T* is the failure time of interest, Δt is a small interval from t to t+Δt, $v\left(t\right)=\left(v_{1}\left(t\right),v_{2}\left(t\right),\cdots ,v_{p}\left(t\right)\right)$ is a set of possibly time-dependent covariates, and $\bar{v}\left(t\right)$ is the history of covariates up to time *t* ($\bar{v}\left(t\right)=\left\{v\left(s\right):0\leq s\leq t\right\})$. The time-dependent Cox PHM is specified as follows [33,34]:(3)$$\lambda (t|\bar{v})=\lambda_{0}\left(t\right)e^{\beta^{\prime }v\left(t\right)}$$
where *λ*_0_ is the baseline hazard function, $\beta^{\prime }=\left(\beta_{1},\beta_{2},\cdots ,\beta_{p}\right)$ is a vector coefficient of regression. In addition, the log-rank rule was used to determine the best split for the node of RSF model. Suppose $\left(T_{1},\delta_{1}\right),\;\left(T_{2},\delta_{2}\right),\cdots ,\left(T_{N},\delta_{N}\right)$ are the survival outcomes corresponding to the N individuals within the node of a tree. Where $\delta_{i}=1$ is event case, and $\delta_{i}=0$ is censored case. Then, the optimized log-rank statistic for the best split of the node on covariate v at split point *c* is represented as [27,35]:(4)$$L\left(v,\;c\right)=\frac{\sum_{i=1}^{f}\left(d_{i1}-n_{i1}\frac{d_{i}}{n_{i}}\right)}{\sqrt{\sum_{i=1}^{f}\frac{n_{i1}}{n_{i}}\left(1-\frac{n_{i1}}{n_{i}}\right)\left(\frac{n_{i}-d_{i}}{n_{i}-1}\right)d_{i}}}$$
where $d_{i1}=\overset{N}{\underset{j=1}{\sum }}I_{\left\{t_{i}\leq T_{j}<t_{i}+dt,\;\delta_{i}=1,\;v_{j}\left(t_{i}\right)\leq c\right\}}$ is the total number of events during the instant interval $\left[t,\;t+dt\right)$ and the covariate $\;v_{j}$ is smaller than c, $d_{i}=\overset{N}{\underset{j=1}{\sum }}I_{\left\{t_{i}\leq T_{j}<t_{i}+dt,\delta_{i}=1\right\}}$ is the total number of events during the instant interval, $n_{i1}=\overset{N}{\underset{j=1}{\sum }}I_{\left\{T_{j}\geq t_{i},\;v_{j}\left(t_{i}\right)\leq c\right\}}$ is the total number of risk at $t_{i}$ and the covariate is smaller than c, and $n_{i}=\overset{N}{\underset{j=1}{\sum }}I_{\left\{T_{j}\geq t_{i}\right\}}$ is the total number of risk at $t_{i},\;i=1,2,\cdots ,f$, and $I_{A}$ is indicator function of set A.

$L\left(v,c\right)$ is a measure of node separation. Herein, the predictor v*** and split value *c** were determined such that $\left|L\left(v^{*},c^{*}\right)\right|\geq \left|L\left(v,c\right)\right|$ and used for all v and *c*. The CHF and survival probability $S\left(t|v\right)=P(T>t|v)$ were calculated at the end node. The KM estimated survival function for any arbitrary time *t* is given by [36]:(5)$$\hat{S}\left(t\right)=\underset{\left\{t_{i}\leq t\right\}}{\prod }\left(1-\frac{d_{i}}{n_{i}}\right)$$

The Cox-based DNNSurv model was used. In DNNSurv model, the function pseudo survival probability for the $j^{th}$ patient was computed by [31,37]:(6)$${\hat{S}}_{j}\left(t\right)=N\hat{S}\left(t\right)-\left(N-1\right){\hat{S}}^{-j}\left(t\right),\;j=1,2,\cdots ,N\;$$

${\hat{S}}^{-j}\left(t\right)$ is the KM estimator using a sample size of N−1, excluding the $j^{th}$ patient. Then, ${\hat{S}}_{j}\left(t\right),\;j=1,2,\cdots ,N$ are used as numeric response variables in the standard regression analysis. Furthermore, the C-index and KM method were used to predict the performance of CKD progression. The C-index is one of the most common discriminatory measures of the survival models and is defined as:(7)$$C-\mathrm{index}=P\left(T_{j_{1}}<\;T_{j_{2}}|\;\eta_{j_{1}}<\eta_{j_{2}}\right),\;for\;j_{1}\neq j_{2},and\;j_{1},j_{2}=1,2,\cdots ,N,\;$$
where $\eta_{j_{1}}=\beta^{\prime }v_{j_{1}},\;\eta_{j_{2}}=\beta^{\prime }v_{j_{2}}\;,\;T_{j_{1}\;},\mathrm{and},\;T_{j_{2}}$ are the predicted marker values and event times, respectively. An estimator of the C-index for survival data is given by [38]:(8)$${\hat{C}}_{surv}=\frac{\sum_{i\neq j}^{N}I_{\left\{T_{j}<T_{i}\right\}}I_{\left\{\eta_{j}<\eta_{i}\right\}}\delta_{j}}{\sum_{i\neq j}^{N}I_{\left\{T_{j}<T_{i}\right\}}\delta_{j}}$$

${\hat{C}}_{surv}$ is a consistent estimator of the C-index where no censoring is present. The C-index depends critically on the variation of the predictors in the cohort study. Similar to the AUROC, a C-index equal to 1 indicates a perfect model prediction, and a C-index of 0.5 represents a random predictor.

The KM method was used as a survival function $\hat{S}\left(t_{i}\right)$ at event time $t_{i}$, as expressed below [39]:(9)$$\hat{S}\left(t_{i}\right)=\overset{i}{\underset{j=1}{\prod }}\frac{n_{j}-d_{j}}{n_{j}}=\hat{S}\left(t_{i-1}\right)*\frac{n_{i}-d_{i}}{n_{i}},\;i=1,2,\cdots ,f.\;$$

We utilized grid search to determine the parameters of the ANN models for hyperparameter tuning while training the model using the defined dataset. Multilayer perceptron was selected to achieve the optimal architecture. Specifically, the model was trained for 10 epochs, consisting of 3 layers with 3 neurons each, a hidden layer of size 3, a batch size of 32, momentum of 0, a learning rate of 0.02, an input layer of size 10 for predictors, an input layer and output layer of size 6 for outcomes, a sigmoid activation function, and an Adam optimizer were used for the network prediction model.

### 2.3. Variables

The baseline characteristics and predictor variables of 497 patients were investigated from the first clinical visit to the endpoint date. Blood tests were performed during the clinical visit for biochemistry testing. In this study, the suitability of categorized and continuous variables of risk factors was assessed and compared for CKD patients with stages 3 to 5. The categorized variables included gender, hypertension, diabetes, and CVD. The continuous variables included age, systolic blood pressure (SBP), diastolic blood pressure (DBP), serum creatinine, HbA1c, PCRln, body mass index (BMI), and eGFR. The eGFR was calculated using the simplified Modification of Diet in Renal Disease equation, which was mentioned in a previous study [40]. All baseline characteristics and predictor variables were obtained from the NHI pre-ESRD patient care and education program administered by the NHI.

## 3. Results

The categorized and continuous variables of CKD patients with stages 3–5 were analyzed. Table 1 depicts the occurrence frequency of categorized variables in CKD patients with stages 3–5, and the number of observations per category is provided. The number of observations for CKD patients in stages 3, 4, and 5 was 935, 416, and 213, respectively. In the present study, a high percentage of hypertension (81%) was observed in stage 5, which increased rapidly and progressively from stages 3–5. The occurrence frequency of diabetes was approximately 50%, with the highest percentage observed in stage 3 (59%). The percentage of CKD patients with stage 3 and CVD was 16%, which decreased from stage 3–5. Additionally, the chi-squared statistical test was used to analyze the difference in CKD patients with and without dialysis. The results showed significant differences in hypertension, diabetes, and CVD between the patients with and without dialysis.

Table 2 shows the clinical characteristics of patient observations with CKD stages 3–5. The serum creatinine and PCRln levels increased progressively from stages 3–5, while the eGFR decreased progressively during the same stages. Furthermore, the F-statistics of clinical characteristics in CKD patients with and without dialysis were analyzed. MANOVA was used to calculate the F-statistics for covariates, allowing the assessment of important risk factors in CKD progression. Significant differences were found in age, serum creatinine, PCRln, and eGFR between the patients with and without dialysis.

The predictive performances of the three models were investigated using the C-index score and KM curves. Table 3 shows the C-index scores of the Cox PHM, RSF, and ANN models obtained through five-fold cross-validation. The average of the C-index scores for RSF is 0.89, with a maximum of 0.95. The average C-index scores of Cox PHM and ANN are 0.71 and 0.72, respectively. Among CKD patients progressing to ESRD within 3 years, RSF demonstrated the best performance. The sensitivity, specificity, and accuracy of RSF are 0.79, 0.88, and 0.86, respectively.

Herein, different cut-off points (0.65, 0.7, and 0.75) of probability in CKD progression were used to determine the sensitivity and specificity of the three models, as shown in Table 4. The results showed that the RSF model had higher sensitivity and specificity compared to the Cox PHM and ANN models. For the RSF model, the sensitivity at the 0.65, 0.70, and 0.75 cut-off points provided values of 0.708, 0.791, and 0.917, respectively. Similarly, the specificity at the same cut-off points provided values of 0.897, 0.880, and 0.794, respectively. Although a high sensitivity (0.917) was observed for the 0.75 cut-off point, a lower specificity was obtained. In addition, the accuracy, precision and F1 score demonstrated the best performance at a cut-off point of 0.70. In the present study, a cut-off point of 0.70 was used to evaluate the CKD patient with and without dialysis in KM curves. The suitable cut-off point could be used to predict the risk factors of CKD–ESRD progression.

Furthermore, the KM curves for nondialysis in CKD patients with different variables (gender, with/without diabetes, stages, and age) were categorized according to the different endpoints. Figure 3 shows the KM curves for the three models in male and female CKD patients with and without dialysis. In Figure 3a, the RSF model demonstrated a higher predicted performance than other models for a male patient without dialysis (the endpoint is 497 days). The predicted probability of the three models is consistently over 0.7, aligning with the actual condition. The three models showed a similar prediction for a male patient without dialysis at an endpoint of 1988 days, as shown in Figure 3b. For a male patient with dialysis, the RSF model showed the best performance within 3 years of progression. The predicted probability of the RSF model is 0.59, which is lower than that of the Cox PHM and ANN models at an endpoint of 1114 days (with a predicted probability of 0.7), as shown in Figure 3c.

In Figure 3d, the RSF model showed a higher performance than other models for a female patient without dialysis (the endpoint is 1420 days). The three models have similar performance to a female patient within less than 1000 days, as shown in Figure 3e. However, the performance of RSF improves after 1000 days, aligning with the actual condition (the endpoint is 835 days). For a female patient with dialysis, a high performance of RSF can be achieved after 625 days (within 2 years), as shown in Figure 3f.

The KM curves of the three models for a CKD patient without diabetes were analyzed. The probability of the three models for patients without dialysis exceeds 0.7 for endpoints of 917 days and 669 days, as shown in Figure 4a,b. For a patient with dialysis, the RSF model demonstrated the best performance for an endpoint of 620 days (within 2 years), as shown in Figure 4c. The predicted probabilities of ANN, Cox PHM, and RSF are 0.8, 0.6, and 0.4, respectively.

In addition, the KM curves of the three models for a CKD patient with diabetes were analyzed. The RSF model shows a higher prediction than other models for endpoints of 469 days and 591 days, as shown in Figure 4d,e. For a patient with dialysis, the best performance of RSF was observed at 768 days (within 3 years), as shown in Figure 4f. The predicted probabilities of ANN, Cox PHM, and RSF are 0.8, 0.8, and 0.2, respectively.

The KM curves of the three models for a CKD patient with stage 3 were analyzed. The ANN model shows a higher predicted performance than the other models at 658 days, as shown in Figure 5a. The three models exhibited similar performance for a patient for 1988 days, as shown in Figure 5b. For a patient on dialysis, a lower predicted performance of RSF was observed. However, the performance of RSF improves after 900 days, as shown in Figure 5c.

Moreover, the KM curves of the three models for a CKD patient with stages 4–5 were analyzed. The RSF showed a higher prediction performance than the other models at 469 days, as shown in Figure 5d. The ANN model showed a higher predicted performance than the other models after 300 days, as shown in Figure 5e. For a patient with dialysis, the best performance of RSF was observed at the endpoint of 768 days (within 3 years), as shown in Figure 5f. The predicted probabilities of ANN, Cox PHM, and RSF are 0.7, 0.7, and 0.2, respectively.

The KM curves of the three models were analyzed for a CKD patient under 80 years old. The RSF showed a higher predicted performance than the other models at 1095 observation days (within 3 years), as shown in Figure 6a,b. For a patient on dialysis, a lower predicted performance of RSF was observed. However, the performance of RSF improves after 900 days, as shown in Figure 6c.

Furthermore, the KM curves of the three models were analyzed for a CKD patient over 81 years old. The RSF model showed a higher prediction than other models after 497 and 532 days, as shown in Figure 6d,e. For a patient with dialysis, the best performance of RSF was observed at the endpoint of 768 days (within 3 years), as shown in Figure 6f.

The RSF was constructed with 100 trees (ntree = 100) and achieved a mean prediction error rate of 0.148. The probability of nondialysis prediction for all CKD patients (N = 429) was analyzed using the three models as shown in Figure 7a. The bold line represents the median, which indicates the average probability of the three models. The interquartile ranges from the bottom to the top of the boxes indicate the 75th and 25th percentiles, respectively. Three models showed great performances for CKD patients who progressed within 3 years. The average probabilities of RSF, Cox PHM, and ANN are 91%, 90%, and 90%, respectively. Moreover, the probability of dialysis prediction for all CKD patients with dialysis (N = 68) was analyzed using the three models, as shown in Figure 7b. The RSF model showed the best performance compared with the Cox PHM and ANN models for patients who progressed within 3 years. The average probability of RSF is 45.38%, which is higher than that of the Cox PHM (14.78%) and ANN models (5.47%). The results demonstrated that the RSF model provides high performance for the prediction of CKD patients with and without dialysis.

According to the C-index values, KM curves, and boxplots for the three models, the best performance was obtained using the RSF model. The VIMP values of risk factors in CKD were analyzed for feature selection and prediction, as shown in Figure 8. Higher VIMP values indicate that the variable may improve the prediction accuracy of the model. The results show that serum creatinine, age, eGFR, and PCRln levels are the most influential features of the RSF model in this study.

## 4. Discussion

This retrospective cohort study included CKD patients who participated in disease management programs for education purposes and the prevention of dialysis in Taiwan. The study demonstrated that PCRln and eGFR showed significant differences among CKD patients with stages 3–5, which were consistent with our previous study [19]. Additionally, hypertension, diabetes, and CVD were significant risk factors among CKD patients with stages 3–5. ESRD among patients was concurrent with many symptoms [41,42,43]. A high ratio (52.3%) of patients with hypertension among 128 ESRD patients was observed by Seyedzadeh [41]. Furthermore, both hypertensive nephrosclerosis and diabetic nephropathy symptoms have a 55% causative role in developing ESRD [42]. Another study shows that diabetes mellitus was the most prevalent comorbidity factor and occurred in 59% of patients, followed by 32.7% with heart disease, among 110 ESRD patients [43]. Thus, hypertension, diabetes, and CVD are important risk factors for ESRD patients. The important categorized and continuous variables can be used as high-risk factors to determine the optimal predictive features for CKD progression.

In the present study, the RSF model demonstrated the best performance, followed by the Cox PHM and ANN models, for the CKD patients who progressed within 3 years. Creatinine, age, GFR, and PCR were found to be correlated with CKD progression in the RSF model. In the Cox PHM model, CVD, along with creatinine, age, and PCR, showed higher predictability for dialysis patients. When applying the same time-dependent design, the RSF model in the present study outperformed the Cox PHM model used in our previous study, even though they utilized different risk factors [19]. However, identifying key features that serve as high-risk factors for determining optimal predictive features is crucial. In past work, de Bruijne et al. showed that the multivariate Cox PHM with time-dependent renal function covariates (serum creatinine, the ratio of serum creatinine, the ratio of serum creatinine at 6 months, and the time elapsed since the last observation) can be used to predict late graft failure in renal transplantation up to 1 year in advance [23]. Moreover, Cox multiple regression with time-dependent covariates has been used for patients with cirrhosis and may be useful for updating the clinical prognosis [44]. Thus, developing a suitable model with important predictors based on time-dependent covariates can increase the efficiency of the clinical strategy intervention.

KM survival curves are commonly used to determine whether risk outcomes vary over time. Recently, the multivariable Cox PHM and KM survival curves were utilized to examine different prognostic factors in 16,752 confirmed cases of COVID-19 [45]. Our findings revealed that the RSF model exhibited higher predictive performance than the Cox PHM and ANN models, regardless of gender and the presence or absence of diabetes. RSF also exhibited the best predictive performance, followed by Cox PHM, for patients with stages 4–5 or ages over 80 years. In contrast, for patients in stage 3 or younger ages, Cox PHM shows a slightly higher prediction performance than RSF among those who eventually received dialysis treatment. Overall, RSF is more suitable than Cox PHM and ANN models for time-dependent risk assessments among CKD patients. A previous study also showed that the performance of RSF (C-index: 0.965) is significantly better than conventional Cox PHM (C-index: 0.766) for 378 patients with kidney transplantation, with RSF particularly useful for intuitive variable selection [24]. Recently, Mondol achieved high accuracy in early CKD prediction using convolutional neural network, ANN, and long short-term memory models [25]. In our recent study, high performance was obtained for predicting CKD progression using random forest methods, with C-indexes of 0.96 within 5 years in the early stage and 0.97 within 1 year in the advanced stage [14]. Although the models proposed by previous studies show high performance, the quality and accuracy of the estimates may vary over time [24,25]. The use of time-independent covariates for individual risk variables often leads to a higher accuracy rate and overestimation percentage. Overestimation can have more significant implications for the care of CKD patients than underestimation. Considering that the trajectories of pathological indicators depend on individual therapy sessions or lifestyle changes, it is crucial to account for the time-dependent influence of covariates on pathological progression.

Data characterization is a common method used to identify important features, which may enhance the credibility of the hypothesis testing in prediction models. Previous studies indicated that the methods of z-score standardization [46], min-max normalization [47], nonlinear transformation [48], and cartesian product [49] can be used for disease prediction with characterized data. In the present study, the time-dependent covariates of renal function were characterized, combining data standardization with the relationship between predictor variables and the observation period. The RSF model, utilizing characterized time-dependent covariates, was used to evaluate CKD progression, which was associated with changes in the pathological records of individual CKD patients with time-varying factors. The RSF model demonstrated a high performance compared to the Cox PHM and ANN models for CKD patients with and without dialysis at stages 4–5. In addition, RSF model could be used to predict the progression for CKD patients with or without diabetes. On the other hand, Cox PHM shows a slightly better prediction performance than RSF for patients with stage 3 or younger ages, among those who were eventually treated with dialysis. The results reveal the potential for useful CKD progression prediction.

Risk prediction based on right-censored survival data by using suitable ML methods has significant implications for CKD patient management. Based on the present results, the proposed method could be used for the risk prediction of CKD progression that allows healthcare professionals to identify individuals who may benefit from early intervention, such as timely referral for transplantation or initiation of dialysis. Additionally, risk prediction models can assist in optimizing healthcare resource allocation by targeting high-risk CKD patients for specific interventions, potentially improving patient clinical outcomes and healthcare efficiency.

## 5. Conclusions

It is crucial to consider the trajectories of pathological indicators in the pathological progression. In this study, being superior to the conventional Cox hazard model, the RSF model was successfully developed using characterized time-dependent covariates, and the results showed good performance for CKD patients at stages 4 and 5 who progress within 3 years. The approach is suitable for personalized prediction of trajectories, even with a relatively small dataset. Creatinine, age, eGFR, and PCR were identified as useful factors for predicting the progression of CKD patients in the RSF model, as indicated by their VIMP values. On the other hand, Cox PHM showed slightly higher prediction performance than RSF for patients with stage 3 or younger ages, among those who eventually required dialysis. The present method could be utilized by physicians and care workers to assess, intervene, and treat CKD progression in a timely manner. However, this study also had several limitations. It was a retrospective cohort study with a relatively small sample size, making it challenging to increase the number of neural network layers for model training. Further studies that analyze clinical pathological records from different hospitals are necessary to ensure unbiased results, thereby increasing the amount of training data to enhance the performance of ANN for predicting progression from early- and advanced-stage CKD. Moreover, the mean age of CKD patients in the study was 80 years, with the maximum age reaching 103 years, which may limit the generalizability of the results to younger patients. Further studies should be verified in CKD patients under the age of 80 to ensure unbiased results and improve the generalization of the model for a wider population.

## Figures and Tables

**Figure 1 biomedicines-11-01664-f001:**
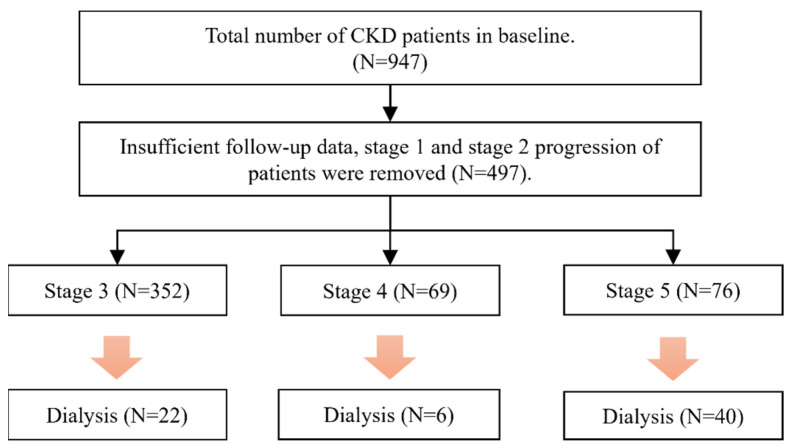
Flow chart for the selection of study subjects.

**Figure 2 biomedicines-11-01664-f002:**
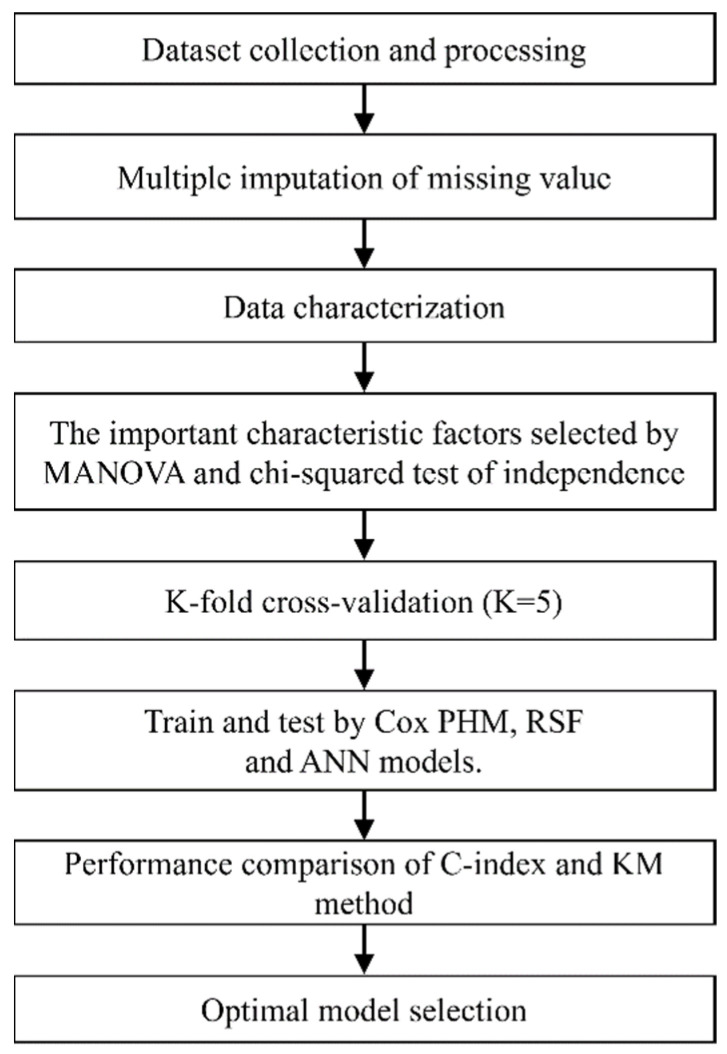
Flow chart of model training and performance evaluation.

**Figure 3 biomedicines-11-01664-f003:**
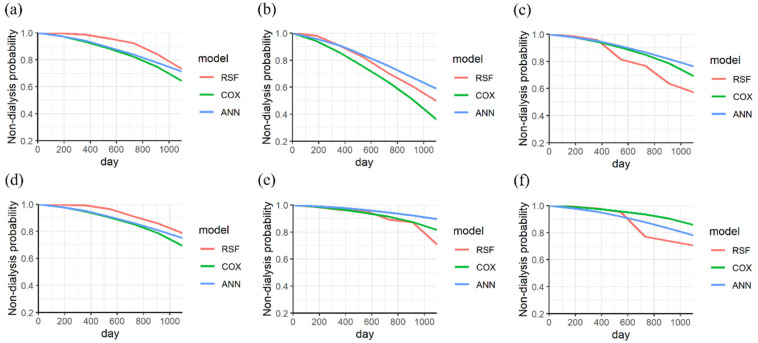
The KM curves of nondialysis in a male CKD patient without dialysis were analyzed at endpoints of (**a**) 497 days and (**b**) 1988 days, (**c**) and that with dialysis at an endpoint of 1114 days. The KM curves of nondialysis in a female CKD patient without dialysis were analyzed at endpoints of (**d**) 1420 days and (**e**) 835 days (**f**), and with dialysis at an endpoint of 1281 days (categorized variable is gender).

**Figure 4 biomedicines-11-01664-f004:**
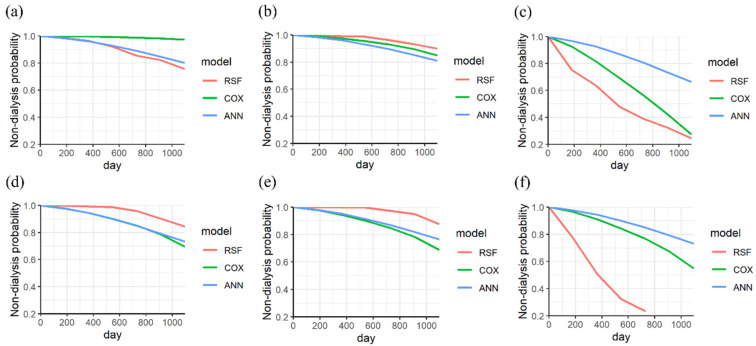
The KM curves of nondialysis in a CKD patient (without diabetes) without dialysis were analyzed at endpoints of (**a**) 917 days and (**b**) 669 days, (**c**) and with dialysis at an endpoint of 620 days. The KM curves of nondialysis in a CKD patient (with diabetes) without dialysis were analyzed at endpoints of (**d**) 469 days and (**e**) 591 days (**f**), and that with dialysis at an endpoint of 768 days.

**Figure 5 biomedicines-11-01664-f005:**
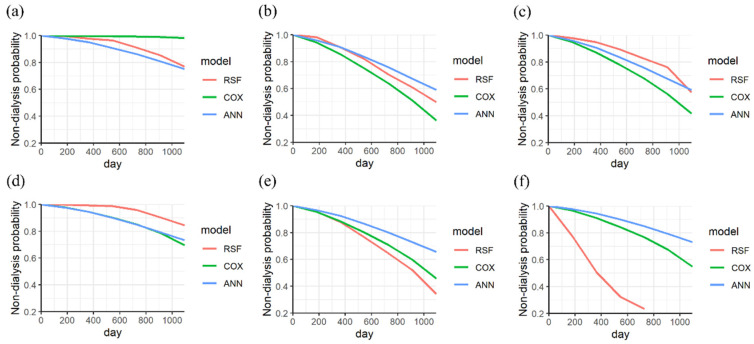
The KM curves of nondialysis in a CKD patient (stage 3) without dialysis were analyzed at endpoints of (**a**) 658 days and (**b**) 1988 days, (**c**) and that with dialysis at an endpoint of 1217 days. The KM curves of nondialysis in a CKD patient (stage 4 to 5) without dialysis were analyzed at endpoints of (**d**) 469 days and (**e**) 532 days (**f**), and that with dialysis at an endpoint of 768 days (categorized variable is the stage of CKD).

**Figure 6 biomedicines-11-01664-f006:**
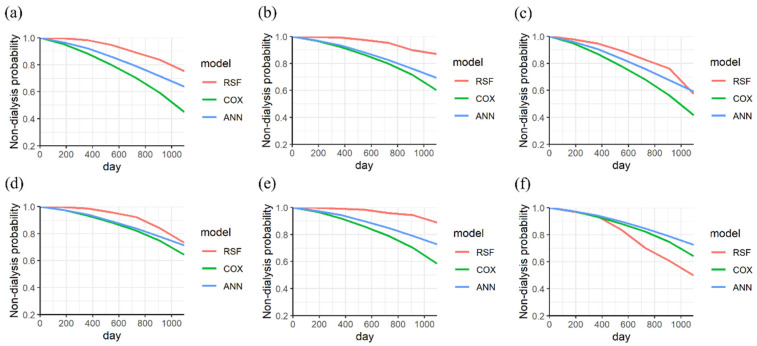
The KM curves of nondialysis in a CKD patient (age ≤ 80) without dialysis were analyzed at endpoints of (**a**) 2022 days and (**b**) 1953 days, (**c**) and that with dialysis at an endpoint of 1217 days. The KM curves of nondialysis in a CKD patient (age ≥ 81) without dialysis were analyzed at endpoints of (**d**) 497 days and (**e**) 532 days (**f**), and that with dialysis at an endpoint of 1958 days (continuous variable is age).

**Figure 7 biomedicines-11-01664-f007:**
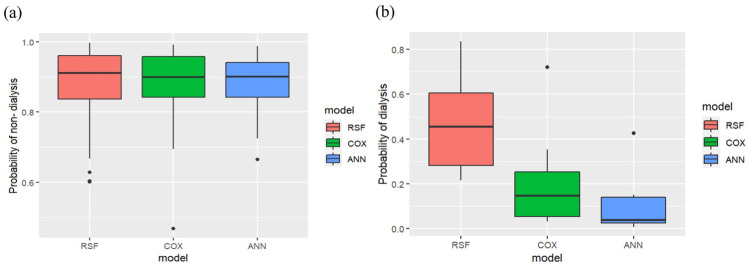
Boxplot distribution analysis for (**a**) probability of nondialysis and (**b**) dialysis with three models in CKD patients at stages 3–5.

**Figure 8 biomedicines-11-01664-f008:**
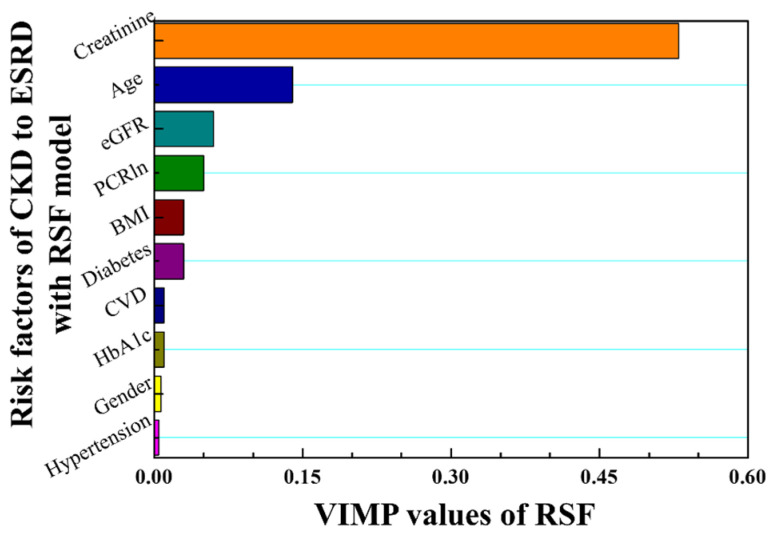
VIMP values are ranked in descending order according to the risk factors of CKD–ESRD progression using the RSF model.

**Table 1 biomedicines-11-01664-t001:** Occurrence frequency of variables categorized by CKD stage.

Variances	Stage 3 (Sub Dataset = 935)	Stage 4 (Sub Dataset = 416)	Stage 5 (Sub Dataset = 213)	Stages 3–5 (Dataset = 1564)			
				With Dialysis (Sub Dataset = 247)	Without Dialysis (Sub Dataset = 1317)	χ^2^	*p* Value
Male	568 (61%)	324 (78%)	126 (59%)	167 (68%)	851 (65%)	0.6944	0.4047
Hypertension	611 (65%)	326 (78%)	175 (81%)	194 (79%)	918 (70%)	14.554	0.00013
Diabetes	552 (59%)	205 (49%)	105 (49%)	164 (66%)	698 (53%)	7.4833	0.0062
CVD	148 (16%)	40 (10%)	3 (1%)	7 (3%)	184 (14%)	23.036	1.58 × 10^−6^

**Table 2 biomedicines-11-01664-t002:** The clinical characteristics are categorized by the CKD stage.

Variances	Stage 3 (Sub Dataset = 935)	Stage 4 (Sub Dataset = 416)	Stage 5 (Sub Dataset = 213)	Stages 3–5 (Dataset = 1564)		
	Mean (SD)	Mean (SD)	Mean (SD)	With Dialysis (Sub Dataset = 247)	Without Dialysis (Sub Dataset = 1317)	F Value
				Mean (SD)	Mean (SD)	
Age	80.57 (11.14)	87.00 (13.11)	76.14 (13.48)	75.85 (13.15)	81.35 (11.77)	43.649 ***
SBP	136.20 (17.92)	136.40 (19.22)	139.60 (19.25)	138.72 (18.65)	136.37 (18.43)	3.391
DBP	73.44 (26.47)	70.34 (12.77)	73.06 (13.21)	73.32 (13.24)	72.42 (23.36)	0.345
Creatinine	1.49 (0.30)	2.56 (0.66)	6.27 (3.55)	4.7 (3.23)	1.99 (1.46)	440.43 ***
HbA1c	6.79 (2.33)	6.47 (1.30)	6.27 (1.17)	6.69 (1.47)	6.61 (2.06)	0.347
PCRln	5.63 (1.49)	6.02 (2.10)	6.78 (2.04)	6.81 (1.93)	5.71 (1.7)	82.182 ***
BMI	26.43 (7.00)	26.66 (15.30)	27.22 (18.16)	26.99 (6.05)	26.52 (12.44)	0.33
eGFR	44.73 (8.18)	22.59 (4.16)	10.15 (3.77)	20.3 (14.19)	36.72 (13.78)	292.58 ***

The significant difference was defined as *** *p* < 0.001.

**Table 3 biomedicines-11-01664-t003:** The C-index scores of five-fold cross-validation in Cox PHM, RSF, and DNNSurv models.

	C-Index					Average
	1	2	3	4	5	
Cox PHM	0.73	0.61	0.73	0.77	0.72	0.71
RSF	0.95	0.91	0.91	0.88	0.81	0.89
ANN	0.75	0.62	0.74	0.73	0.73	0.72

**Table 4 biomedicines-11-01664-t004:** The sensitivity and specificity of three models at different cut points.

Model	Cut off Point	Sensitivity	Specificity	Accuracy	Precision	F1 Score
RSF	0.65	0.708	0.897	0.867	0.567	0.630
	0.70	0.791	0.880	0.897	0.792	0.791
	0.75	0.917	0.794	0.813	0.458	0.610
Cox PHM	0.65	0.083	0.929	0.793	0.181	0.114
	0.70	0.125	0.920	0.793	0.231	0.162
	0.75	0.125	0.897	0.920	0.187	0.150
ANN	0.65	0.000	0.984	0.827	0.000	0.000
	0.70	0.000	0.952	0.800	0.000	0.000
	0.75	0.041	0.913	0.773	0.083	0.055

## Data Availability

The datasets generated and analyzed during the current study are not publicly available because of privacy and ethical restrictions, but are available from the corresponding author upon reasonable request.

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
