# Peer review of "Improved Survival Analyses Based on Characterized Time-Dependent Covariates to Predict Individual Chronic Kidney Disease Progression"

_biomedicines, 2023, doi:10.3390/biomedicines11061664_

Round 1
Reviewer 1 Report
1. Problem statements and research gaps that lead to current research works are not clear.
2. Please highlight the novelty and contributions of current researches to highlight the significance of current study.
3. Why only three predictive models (e.g., Cox PHM, RSF and ANN) are considered in this study? What are the justifications of selecting these predictive models instead of others?
4. The meaning of the time-dependent variability analysis in Eq. (1) should be explained further for better clarity.
5. How to determine the value of risk factor v mentioned in Eq. (1)?
6. More sophisticated machine learning models (e.g., XGBoost) should be used for the more thorough comparative studies.
7. What are the future works that can be extended from current study?
English quality is acceptable.
Author Response
Please see the notes file.

Reviewer 2 Report
Review of the article "Improved survival analyses based on characterized time-dependent covariates to predict individual chronic kidney disease progression"
The article is valuable with clearly visible aim and novelties.
Points to correct the article:
1. There is in the Abstract (line 14): "who were treated at a hospital from January 2006 to 2019 in Taiwan."
I propose to change to "who were treated at the ward of Taipei Veterans General Hospital from January 2006 to 2019 in Taiwan."
2. There is in the Discussion section (lines 435 - 437): "In addition, RSF model demonstrated a high performance compared with Cox PHM and ANN models for CKD patients with and without dialysis at stage 3 to 5."
but in the lines 408 - 410 is "In contrast, for patients with stage 3 or younger ages, Cox PHM shows a slightly higher prediction performance than RSF among those who were eventually treated with dialysis.",
so, for patients with stage 3, RSF model did not demonstrate a high performance compared with Cox PHM for CKD patients with and without dialysis as it is in the lines 435 - 437.
This mistake should be corrected.
3. The performance of an artificial neural network depends on the parameters of this network and the way it has been taught.
For these reasons, please add information about the parameters of the neural network you used, among others: the number of layers, the number of neurons in the input layer, the number of neurons in the hidden layer(s), the number of neurons in the output layer, learning rate, momentum, the number of patterns, teaching error and in what way this ANN has been taught.
Please also add more information about the implementation of this neural network.
4. In my opinion, the following part of the work should be moved to the Conclusions section:
"This study also had several limitations. This retrospective cohort study consisted of a relatively small sample, so it can be difficult to increase the number of neural network layers for model training. Moreover, the mean age of CKD patients was 80 and the maximum age was 103 years old, thus the present results may not be suitable for a younger range of patients. The generalization of the model for the wider population may be limited. Future studies that perform an analysis of clinical pathological records from different hospitals are necessary to ensure that the results are unbiased."
5. The Conclusions section is too short and written in a very general way (i.e. it contains very general information). The Conclusions section needs to be extended, moreover, concrete conclusions from the work should be presented in this section, supported by the results obtained in the work.
6. The article has been carefully written, although linguistic mistakes require correction.
The article has been carefully written, although linguistic mistakes require correction.
Author Response
Please see the notes file.

Reviewer 3 Report
The manuscript entitled “Improved survival analyses based on characterized time-dependent covariates to predict individual chronic kidney disease progression “has been investigated in detail. The topic addressed in the manuscript is interesting but there are many questions which should be dressed by authors:
1. Keywords should be written in alphabetical order.
2. More metric may be used for comparison of simulation results.
3. Some mathematical notations are not rigorous enough to correctly understand the contents of the paper. The authors are requested to recheck all the definition of variables and further clarify these equations. Definitions of all variables and their intervals should be given.
4. What are the other possible methodologies that can be used to achieve your objective in relation in this work?
5. Comparison the performance of the proposed method is poor. For example, there are a lot of different ANNs.
6. Which kind of ANN has been used? And why? Please explain in the manuscript by detail.
7. How the best structure for selected ANN model has been obtained? For example, if RBF has been selected, how the best spread was obtained? if MLP has been selected, how was the best architecture (number of neurons, epochs, kind of activation function and etc.) of network obtained? Please discuss it in the manuscript precisely.
8. Please add the results of presented models in the manuscript. For example regression diagrams, error versus number of neurons, training and test errors and etc.
9. Why the overfitting problem has not been occurred for different models?
10. It will be helpful to the readers if some discussions about insight of the main results are added as Remarks.
This study may be consider for publication if it is addressed in the specified problems.
The linguistic quality needs improvement.
Author Response
Please see the notes file.

Round 2
Reviewer 1 Report
Authors have addressed all my comments raised in previous review cycle. No further comments.
Quality of English Language is good.
Reviewer 2 Report
I am satisfied with the additions and corrections made by the Authors in accordance with my comments. Now the quality of the article is better and I recommend it for publication in Biomedicines.
Reviewer 3 Report
My recommendation is "Accept in present form".